# Sol-Gel Derived Hydroxyapatite Coatings for Titanium Implants: A Review

**DOI:** 10.3390/bioengineering7040127

**Published:** 2020-10-14

**Authors:** Alaa Jaafar, Christine Hecker, Pál Árki, Yvonne Joseph

**Affiliations:** Institute of Electronics and Sensor Materials, TU Bergakademie Freiberg, 09599 Freiberg, Germany; Alaa.Jaafar@Doktorand.tu-freiberg.de (A.J.); Christine.Hecker@esm.tu-freiberg.de (C.H.); Yvonne.Joseph@esm.tu-freiberg.de (Y.J.)

**Keywords:** hydroxyapatite, sol-gel, titanium alloy, biocompatibility, implant

## Abstract

With the growing demands for bone implant therapy, titanium (Ti) and its alloys are considered as appropriate choices for the load-bearing bone implant substitutes. However, the interaction of bare Ti-based implants with the tissues is critical to the success of the implants for long-term stability. Thus, surface modifications of Ti implants with biocompatible hydroxyapatite (HAp) coatings before implantation is important and gained interest. Sol-gel is a potential technique for deposition the biocompatible HAp and has many advantages over other methods. Therefore, this review strives to provide widespread overview on the recent development of sol-gel HAp deposition on Ti. This study shows that sol-gel technique was able to produce uniform and homogenous HAp coatings and identified the role of surface pretreatment of Ti substrate, optimizing the sol-gel parameters, substitution, and reinforcement of HAp on improving the coating properties. Critical factors that influence on the characteristics of the deposited sol-gel HAp films as corrosion resistance, adhesion to substrate, bioactivity, morphological, and structural properties are discussed. The review also highlights the critical issues, the most significant challenges, and the areas requiring further research.

## 1. Introduction

Recently, over 90% of the elderly populations of the world are suffering from bone-related trauma, such as osteoporosis, bone cancers, rheumatoid arthritis, or accidents, which require replacement procedures for the spinal, hip, and knee with biomaterials [1]. Thus, there is a great demand for appropriate biomaterial with excellent combination of low elastic modulus and high strength, high wear and fatigue resistance, superior corrosion resistance in the body environment, high longevity, biocompatible, and without cytotoxicity [2]. Various artificial bone materials have been applied to substitute diseased bones, as ceramics, metals, polymers, and their composites. Compared to polymers and ceramics, metals and its alloys consider the most appropriate choice for the replacement of the damaged load-bearing bones due to their high mechanical properties, where, according to some statistics made, over 70% of the manufactured implant devices are made of metallic biomaterials [3,4].

The most promising types of the implantable metallic biomaterials are titanium and its alloys, stainless steel and cobalt chromium alloys. Among these materials, titanium and its alloys attracted high importance due to its highest biocompatibility among other alloys, alongside with its combination of high mechanical properties, excellent corrosion resistance high strength and relatively low weight [4,5]. Titanium alloys are present in several phases as α, near-α, α + β, metastable β and stable β. The Incorporation of alloying elements could work as stabilizer to α phase (i.e., O, Hf, Ta, N, Al, and C) or to β phase (i.e., H, Nb, V, Si, Co, Fe, Mo, Mn, Mo, and Ni) or neutral elements (i.e., Zr) [6]. Titanium alloys with α and near-α phase have high corrosion resistance, but their mechanical properties are limited. On the other hand, Ti alloys with β phase have poor corrosion resistance compared to α alloys but can be shaped at relative low temperatures due to their body-centered cubic crystal structure. So, coupling the benefits of α and β phases can produce Ti alloys ideal for orthopedic implants [6,7,8]. The most common grade of biomedical titanium alloys that utilized in bone replacement is grade 5 which is referred as Ti6Al4V with composition of 6% Al and 4% V, addition of these elements can significantly improve the mechanical strength of the titanium alloy due to their role of acting as stabilizers of the α + β phases of titanium [9]. However, vanadium and aluminum are both not biocompatible and toxic, and that was the driving force to develop new generations of biomedical titanium alloys with toxic-free components that are relying on molybdenum, iron and niobium as β phase stabilizer and zirconium, hafnium and tantalum as α phase stabilizer [10].

The Ti-based implants still have many limitations to be implanted inside the human body, the high elastic modulus of the Ti compared to the bone can induce stress shielding that leads to the implant failure due to the limitation of restructuring and resorption of the bone [9]. In addition, the poor tribological behavior of Ti due to its high friction coefficient could cause severe adhesive wear and consequently generate debris in the bloodstream resulting in bone resorption. An inflammation in the surrounding tissues leading to implant loosening, hence, a second surgery is needed to implant a new one [2,11,12]. Furthermore, these materials present damaging influences on the living organisms when used as implant due to their low corrosion resistance in the body environment, this drawback is an important factor that limits the biocompatibility of these materials by releasing toxic and undesirable products in the body fluid [13,14]. Moreover, Ti and its alloys are bio-inert materials, i.e., they do not induce allergic reactions. However, these materials recognized as a foreign body and isolated in a fibrotic capsule, therefore, the osseointegration is hampered and regeneration of natural bone is impossible [9,15]. All these limitations were the driving force towards developing methods for functionalization, the surfaces of the Ti implant by modifying the morphology, composition and structure of their surfaces leaving intact the mechanical properties [16].

Surface modification is one of the easiest ways to overcome the limitations of the Ti implants and achieve a better biological outcome by promoting the bioactivity of the surface, prevent implant-related infections and eliminate or control the degradation rate while the desirable bulk attributes of the materials are retained [17,18,19]. Deposition of bioceramic coatings on the metal-implant surface is one of the most common surface modification processes [20]. Despite the poor mechanical and tribological properties of these materials, they can enhance the biological fixation between the Ti implants and the bone leading to increase the clinical success rate in the long-term compared to the uncoated Ti implants [16].

Bioceramic coatings are classified into two main categories: Bioinert and bioactive coatings. The bioinert ceramic coatings, such as alumina and zirconia, have a good biocompatibility and higher mechanical properties compared to bioactive ceramics. However, the high brittleness, high elastic modulus and the poor interaction ability of these materials with the surrounding tissues limiting their application in this field. On the other hand, the bioactive ceramic coatings, such as calcium phosphate and bioglass, are more extensively used to treat the Ti implant due to their abilities to enhance the adhesion between the implant and the bone via their guaranteed interaction with the body.

Recently, the bioactive calcium phosphate has gained widespread attention in the medical coating technology due to their important role in reducing the healing time by promoting a strong connection between the surrounding tissue and the implant. There are different types of calcium phosphate with bioactive features in many crystalline phases such as hydroxyapatite (HAp), Tricalcium phosphate (TCP), Whitlockite (WH) etc. [21]. Among these types, hydroxyapatite (Ca_10_(PO_4_)_6_(OH)_2_) have received enormous considerations as a coatings on the metallic implants surface [22] due to its crystallographical, chemical and mineralogical composition that resemble the human bone [23]. Deposition of HAp films on metallic-based implants advanced the orthopedic applications by production of biomaterials that combine the appropriate mechanical properties and the high biocompatibility and bioactivity of the surface [24]. The favorable in-vivo behavior of the HAp films led to promote the bone bonding ability and prolongs the metallic prosthesis lifetime. In addition, HAp coating enhances the corrosion resistance of the biomaterials by acting as a barrier against the releasing metal ions [25,26]. Yang et al. [27] reported that deposition of HAp film develop the clinical success with a less than 2% failure rate. However, for the HAp coating to be effective in its surface function, it should have sufficient bonding strength with the surface to connect the implant with the bone tissues, outstanding stability and dissolution resistance and suitable morphology to increase the contact area with the bone [28]. The minimal requirements for HAp coatings have been described in the ISO standards as well as the Food and Drug Administration (FDA) guidelines as demonstrated in Table 1 [29,30]. Hence, deposition of high quality HAp coating that could satisfy the standards is an essential challenge.

## 2. Deposition Techniques of Hydroxyapatite

There are numerous potential coating techniques with capability to deposit HAp films on metallic implants, the most frequently applied techniques are: sol–gel coating [9,31,32,33,34], plasma spraying [35,36,37], biomimetic deposition [38], electrochemical deposition [39,40], and electrophoretic deposition [41,42]. Table 2 summaries the benefits and drawbacks of each technique.

Among the techniques listed, plasma spraying is the only method commercially approved by the FDA for biomedical coatings on implants [43]. High biocompatibility and bioactivity, good adhesion, suitable dissolution resistance, and many other benefits could be attained by deposition the HAp via plasma spraying as described previously by Heimann [44]. However, this technique has some main drawbacks, the high temperature of this method may adversely affect the final properties of the Ti substrate and HAp coatings and noticeably effect on the benefits of the produced coating. For instance, the high processing temperature could decompose the HAp to CaO and Ca_3_PO_4_, these compounds are not biocompatible and degrade after a short time of implantation [45]. In addition, the relatively high thickness of the plasma spraying HAp coating (>30 µm) consider a drawback and can cause a delamination of the coating layer after implantation [46,47].

## 3. Sol-Gel Deposition of Hydroxyapatite

Sol-gel coating has many advantages over the other techniques listed, as this technique is perform at low processing temperature [64], with low cost [65], as well as producing coatings with high purity and homogeneity [66]. In addition, this technique is able to produce uniform and intimate mixtures of different colloidal oxides on a molecular level and the resulting gel can be shaped easily, also, the sol-gel method could enable the control of the chemical composition on a molecular level, so very small quantities of different components can be applied to the sol and dispersed uniformly [67,68].

The chemistry of sol-gel coating technique has several stages including, (1): formation of colloidal solution by hydrolysis and partial condensation of molecular precursors, (2): formation of gel material with three-dimensional network by condensation of the sol particles, (3): aging and (4): drying [69]. The sol-gel chemistry can be simplified the following reactions [69]:Hydrolysis: –M–OR + H_2_O → –MOH + ROH(1)
Condensation: –M–OH + XO–M– → –M–O–M– + XOH(2)
where: M = metal and X = H or R (alkyl group).

### 3.1. Preparation of the Hydroxyapatite Sol

For manufacturing of sol-gel HAp coating, combination of calcium and phosphorus precursors are used for preparation of sol with addition of solvents, often ethanol and water [70]. The chemical nature of Ca and P precursors play a vital role to produce the hydroxyapatite [71,72]. An extensive review on fabrication sol-gel derived calcium-phosphate powders using different precursors have been done previously by Ishikawa et al. [73]. Table 3 shows the most frequent precursors and solvents used for preparation of hydroxyapatite. Generally, the selected phosphorus and calcium precursors dissolve in solvents separately, then, mix together in a dropwise approach with controlling the molar ratio [74,75]. The molar ratio of Ca/P in the bone apatite is detected to be 1.67, which is necessary to synthesis HAp with a ratio as close as possible to 1.67 to simulate the body environment [76]. The mixture is then agitated at various temperatures, and solvents are evaporated off until a more viscous sol is obtained [77].

### 3.2. Deposition Approaches of Hydroxyapatite Sol

Several approaches can be utilized to deposit the produced HAp sol on the Ti substrate, the most common are dip coating and spin coating. In dip coating method, the Ti substrate is immersed and withdrawn vertically at a constant speed from the desired coating sol [69]. In some researches, stirring after immersion is carried out in order to achieve homogeneity in the entire sol volume [76]. Withdrawal speed, viscosity of sol, time of immersing, and number of dips are the factors controlling the film thickness [69,98]. In spin coating, drops of sol are dispensed on a substrate surface and the substrate is spun at a high speed afterwards. Sol viscosity, spinning velocity, and surface tension are the parameters controlling on and thickness of the film [99]. Commonly, dip coating used to coat complex shapes, while spin coating is preferred with flat surfaces [100]. Figure 1 shows a schematic rendering of HAp preparation and the stages of dip and spin coating.

### 3.3. Limitations of Sol-Gel Derived Hydroxyapatite Coatings

In spite of biological performance enhancements that HAp can provide to the Ti surface, there are still many concerns about the use of HAp coatings, especially with regard to long-term reliability [101]. Implant retrieval studies have shown remarkable degradation of HAp coatings over short implantation time due to the poor adhesion of HAp-Ti interface [101,102]. The mismatch in coefficient of thermal expansion (CTE) between HAp and Ti surface is the main reason for the poor adhesion [86,103,104]. A detailed explanation about the mismatch in CTE and the residual stresses formed at HAp-Ti interface is made previously by Carradò and Viart [105]. However, the absence of chemical bonding between and the HAp and Ti surface can be also a reason for this drawback [48].

The brittleness of hydroxyapatite and its low tensile strength and fracture toughness are often resulting premature fracture of the coating layer [81,106]. Moreover, the low mechanical properties of hydroxyapatite lead to form cracks and expose the underlying metallic implant body to corrosion and infection, which limits its application as a coating [107,108]. In addition, many in-vitro examinations have demonstrated that pure HAp coatings are suffering of high dissolution rate in the simulated body fluid (SBF) which is lead to disintegration of the HAp coatings and hinder the fixation of implant to the host tissue [109,110]. Furthermore, amorphous or poorly crystalline HAp films are usually produced using sol-gel technique [23], while highly crystalline HAp coatings are preferable, Xue et al. [111] illustrated the importance of high crystallinity in reduction of in-vivo dissolution of HAp films. It is also noticed that higher content of crystalline HAp increase the adhesion strength between the HAp and substrate [112]. Nevertheless, hydroxyapatite readily absorbs the proteins and organic substances in the body, which, in turn, absorb and replicate the bacteria, leading to infections [113].

To tackle these shortcomings, three main strategies have been applied: First, the sol-gel coating parameters were optimized; Second, the interface between the Ti implant and the HAp coating was engineered; And third, the HAp was reinforced by various filler materials to form composites or by ion exchange to achieve designed materials properties. This review presents these strategies in detail and discuss their effect on the quality of the coatings.

## 4. Optimizing the Sol-Gel Processing Parameters

Synthesis and deposition of sol-gel derived HAp on Ti implants requires extremely stringent processing parameters such as the chemical compositions of the precursors, pH value, temperature and time of sol preparation, sol viscosity, and sintering duration and temperature. Major resulting issues include the purity, crystallinity, stability, adhesion strength, and biocompatibility of the HAp coatings are influenced by these parameters.

The first stage for synthesis of sol-gel hydroxyapatite is to mix solutions of various phosphorus and calcium precursors. The condition of the mixture including the pH value, temperature, and duration of mixing have crucial role on the formation of HAp. To understand the impact of each factor, Sadat-Shojai et al. [114] have investigated the optimum conditions for HAp synthesis with aid of statistical design of experiment. The investigation revealed the domination of the mixture pH value on the nucleation of HAp phase. In harsh acidic media (pH = 4), only pure dicalcium phosphate dehydrate (DCPD) and dicalcium phosphate anhydrous (DCPA) are nucleated. However, increasing the temperature and time of mixing up to 200 °C and 100 min could be attributed to HAp formation. With original pH value of the mixture (pH = 5.6), the HAp phase could form with the presence of small amount of octacalcium phosphate (OCP) and DCPA. While the alkaline media (pH = 9) shows the nucleation of pure HAp irrespective to the temperature. Despite the nonparticipation of the temperature on the purity on HAp at alkaline media, it is reported that HAp crystallinity is significantly increase with high mixing temperatures.

Aging time of the sol is another key factor effecting the final composition of the sol-gel HAp coating because it is important to allow complete formation and homogenization of a stoichiometric composition. Chai et al. [66] discussed this subject and made an attempt to estimate the critical aging time that alkoxide-based precursors are required to produce monophasic HAp by applying various aging times ranging from 15 min to several days. Based on Differential Thermal Analysis (DTA) and XRD analysis (Figure 2), other phases including CaO are observed in the HAp coatings with lower than 24 h aging time suggesting that period greater than 24 h is important to produce monophasic HAp.

Kang et al. [87] studied the influence of sol viscosity and sintering temperature on the morphology and crystallographic properties of hydroxyapatite coated on anodized Ti substrate. The results indicated that the sol viscosity has no influence on the crystallization and formation mechanism of the HAp. However, lowering the viscosity attribute to decrease the thickness of the deposited film. Kim et al. [64] developed the sol-gel HAp coating on Ti substrate by addition of ammonium hydroxide (NH_4_OH) with small amounts (1–5 vol.%) to the ethanol-based HAp solution. The viscosity and pH of the HAp sol found to be increased with increasing the NH_4_OH concentration that contributed to improve the polymerization and gelation of the sol, which, in turn, shortened the aging time required for crystallization of the coating.

Sintering temperature in the final heat treatment step of sol-gel coating is critical parameter in determination the quality of the coating. The generation of crystalline phase, densification of the gel film and eliminating the porosity are occurring during this process, thus, controlling this parameter is essential [115,116]. Gross et al. [22] described the essential role of the sintering temperature for production of sol-gel HAp coatings on titanium substrates. In their study, X-ray diffraction (XRD) analysis have indicated the starting of uncontrolled oxidation and phase transformations of Ti substrate for the specimens heat treated higher than 800 °C. Thus, it was suggested that sintering temperature for the Ti implants coated with HAp should be around 800 °C to decrease the oxidation occurrence and reduce the possibility of phase transformations in the Ti substrates. However, densification of the HAp coatings could be achieved with a longer sintering duration at 800 °C. Similarly, Jafari et al. [117] reported drawback in the morphological properties and formation of cracks on the surface of the sol-gel hydroxyapatite film deposited on Ti6Al4V after increasing the sintering temperature up to 800 °C (Figure 3c). This drawback might be attributed to the phase transformation of the Ti alloy at that temperature. From the observations, porous HAp coating surface is found at 600 °C (Figure 3a), and rising the sintering temperature to 700 °C considered beneficiary to decrease the porosity and produce denser HAp structure (Figure 3b). Furthermore, Jafari et al. [31] used sol-gel dip coating method to deposit nano-hydroxyapatite on Ti alloy with different sintering conditions. The results of structure and surface morphology analysis found that increasing the sintering temperature up to 700 °C and extending the time to 30 min can be attributed to improve the integrity of crystallized HAp films, reduces coating defects, and enhances the mechanical properties of the deposited film. The XRD analysis revealed a development in HAp crystallization with increasing the sintering temperature. Introduction of HAp coating have improved the corrosion resistance of the Ti alloy by 48%, while sintering the coating layer at 700 °C has significantly improved the corrosion resistance by 17 times compared to the pure alloy and presented the highest corrosion behavior.

Contrary to the studies [22,31,117], some other studies adopted higher sintering temperature in their investigations. Stankevičiūtė et al. [78] studied the influence of the dip-coating conditions and heating time on the structure and morphology of sol-gel HAp coating sintered at 1000 °C. The XRD analysis revealed the formation of crystalline HAp, however, formation of crystalline titanium dioxide (TiO_2_) is also observed due to the high sintering temperature. Identifying the formed TiO_2_ layer is performed by sintering the uncoated pure Ti repeatedly, which observed an increase in the size of TiO_2_ crystallites with increasing the duration of annealing. However, the number of coating layers has no impact on the coating crystallization. The SEM micrographs showed smaller and homogeneously distributed spherical particles with increasing the amount of layers up to 15 layers. Yavuz et al. [84] studied the effects of sintering temperature and surface pretreatments (acid passivation and hydrogen sputtering) on the bonding strength of sol-gel HAp coating on commercial pure titanium (cp-Ti). The scratch test result reported an enhancement in adhesion with increasing sintering temperature up to 900 °C. Besides, the hydrogen sputtering showed a better capability to improve the adhesive strength compared to the acid passivation.

The influence of heating rate in the post-heat treatment on the HAp coating’s surface morphology, crystal structure and adhesive strength has been discussed by Wang et al. [118]. They reported that the synthesis of HAp needs sufficient time and the rapid heat treatment negatively influences the coating structure by accelerating the decomposition of HAp as demonstrated in Figure 4. In addition, the high heating rate (100 °C/min) found to induce higher thermal stresses and resulting lower bonding strength compare to slow heating rate (20 °C/min) in the scratch test. In another study, Kim et al. [119], observed formation of uniform and smooth HAp film using a heat rate of 1 °C/min, while increasing the heating rate up to 50 °C/min could create irregular films with 4–6 times higher roughness compared to the low heating rate. In addition, they found that the dissolution rate of the rougher surface in simulated body fluid (SBF) was slightly higher than that of smooth surface due to the higher exposed surface area that interact with medium.

## 5. Engineering of the Interface

Pretreatment of the Ti surface by creation an intermediate layer between the Ti-substrate and HAp coating is a practical way to overcome many challenges associated with deposition of HAp by sol-gel. These interlayers reduce the mismatch of coefficients of thermal expansion between HAp and substrates, thus increasing the coatings bonding strength without affecting biocompatibility [43]. Also, they can slow down the fast cooling rates, decrease the thermal decomposition of HAp, and enhance the crystallinity of the coatings [45]. In addition, the interlayers can satisfactorily give rise to the mechanical, biological, and corrosion performance of the substrates. For instance, Narayan [120] have demonstrated in his report that the deposition of interlayer not only minimize the probability of inducing corrosion of HAp in body fluids, but also improves implant fixation due to the minimal fibrous capsule around the implant.

### 5.1. TiO_2_ Interlayer Prepared by Sol-Gel

Titania (TiO_2_) has always been a favorable choice as an interlayer between the titanium substrate and HAp coating due to its high mechanical integrity with HAp and its chemical similarity to both HAp and titanium [121]. In addition, TiO_2_ has outstanding tribological properties, antibacterial activity, and its biocompatibility has been confirmed by in-vitro and in-vivo biological studies [122,123,124]. Furthermore, the mismatch of the thermal expansion coefficients between the HAp and Ti can be reduced by introducing TiO_2_ interlayer. Kim et al. [125] have prepared a double layer of TiO_2_ and HAp on cp-Ti substrate using sol-gel spin coating. From the observation, a tight bonding between TiO_2_-Ti and TiO_2_-HAp, and the bonding strength of the TiO_2_-HAp bilayer was improved by 60% compared to the pure HAp coating. In addition, significant improvement of corrosion resistance is noted after applying TiO_2_ coating, as the corrosion current density of Ti samples (~9.5 × 10^−7^ A/cm^2^) reduced to (~4.2 × 10^−5^ A/cm^2^) with the coated samples. In another study, Balakrishnan et al. [126] found that the role of TiO_2_ interlayer in reducing the mismatch of the thermal expansion coefficient between the HAp layer and Ti6Al4V substrate can effectively produce crack-free HAp coating and reduce the delamination in the interface. The bonding strength in this study promoted from 15.8 ± 7 MPa to 40.3 ± 3 MPa after introduction of TiO_2_.

Azari et al. [33] fabricated functionally graded of HAp-TiO_2_ coating with three layers (100% TiO_2_, 50% TiO_2_ −50% HAp, 100% HAp) on the Ti6Al4V alloy using sol-gel spin coating. Cross-sectional SEM images as demonstrated in Figure 5 revealed that introducing the TiO_2_ layer and grading its composition with HAp improves the bonding strength and cohesion of the coating and reduce the interface delamination compared to single HAp coating. Xu et al. [127] deposited a double layer of HAp/TiO_2_ on the pure titanium using sol-gel spin coating. The SEM observations found no delamination in the interfaces between the Ti-TiO_2_ and TiO_2_-HAp layers, suggesting that the adhesive strength is very strong. Hussein and Mohammed [128] investigated the improvement of corrosion characteristics of Ti alloy after deposition of TiO_2_ layer and TiO_2_-HAp multilayer. The results of electrochemical measurements revealed a superior anti-corrosion property of the bilayer coating. The corrosion resistance of the bilayer is enhanced about 32% compared to the TiO_2_ coated specimen and 63% compared to the uncoated specimen.

### 5.2. TiO_2_ Interlayer Prepared by Anodization

Other researchers suggested creating the TiO_2_ layer by oxidizing the substrate surface instead of coating. Anodization as a pretreatment process has been suggested as an effective way to create TiO_2_ layer on the Ti surface. In this method, the titanium oxide is formed by moving the titanium ions through the surface oxide layer and react with the oxide ions in the solution. This method allows the formation of an oxide layer whose thickness can be accurately controlled by a combination of type of electrolyte, electric current, and time to achieve film thicknesses of Angstroms order (10^−10^ m) [129]. In this regard, Roest et al. [130] anodized commercial cp-Ti and Ti6Al4V using different voltages (25, 50 and 75 V) and applied the HAp coating by sol-gel spin coating. Based on XRD result, small amount of rutile phase were observed at 25 V anodization voltages, while only anatase phase is detected at higher voltages. An improvement in wettability on both cp-Ti and Ti6Al4V was reported by increasing the voltages up to 50 V due to the reduction of contact angle associated with increment in surface roughness. The tensile test result revealed improvement in the adhesive bonding of HAp to the anodized substrates compared to the as-polished substrates, and it was more noticeable on the Ti6Al4V. A rise in adhesive strength can be also noticed with increasing the anodization voltage up to 50 V.

Ji et al. [86] applied sol-gel derived hydroxyapatite and hydroxyapatite-carbon nanotubes (CNTs) composite coating on anodized and non-anodized Ti substrates. XRD results showed an amorphous phase of TiO_2_ before the heat treatment, while anatase and rutile phases were produced at 450 °C and 700 °C sintering temperature, respectively. Higher rate of HAp crystallization resulted with homogeneous dispersion of CNTs. They observed that both HAp and HAp-CNT coatings bonded more tightly on the TiO_2_ compared to the Ti substrate. The improvement in adhesive strength is also reported by Robertson et al. [93] after introduction of TiO_2_ as an intermediary layer on the Ti6Al4V surface by anodization process. They observed in the results of tensile test, an improvement in adhesive strength nearly to 38% after introduction of TiO_2_ compared to bare HAp coating. The anodized TiO_2_ layer enhances the corrosion resistance by being as a barrier to embed the releasing of metallic ions. This is also confirmed by Kang et al. [131] who reported an enhancement in electrochemical properties by lowering in corrosion current density and development in corrosion resistance after anodizing the surface of cp-Ti coated by sol-gel hydroxyapatite.

### 5.3. Other Types of Interlayer

Creation of titanium boride interlayer in the Ti-HAp interface by boronisation of Ti surface has been suggested by Esfahani et al. [90]. They found that boronising the Ti resulted in two layers of titanium boride, an inner layer of TiB and an outer layer of TiB_2_. Remarkable improvement in mechanical properties is observed after deposition the TiB and TiB_2_, micro hardness observed to be much higher on the boronised samples (~1750 HV) compared to the bare titanium substrate (180–250 HV). The wear resistance of the treated samples found to be eight times higher than untreated specimens. The result of the pull off test reported remarkable improvement in bonding strength of HAp coating from 3 to 15 MPa after introducing the titanium boride intermediate film. In another study, Kazemi et al. [132] have deposited titanium nitride (TiN) as interlayer by plasma-assisted chemical vapor deposition (PACVD) with sol-gel HAp coating on Ti alloy. Introduction of TiN layer had improved the surface roughness of the HAp coating to bone-like structure, which, in turn, promoted the apatite formation ability compared to single HAp film. In addition, the corrosion resistance of the multilayer is remarkably improved, where the corrosion current density of the single HAp coating (0.12 μA/cm^2^) is reduced by 50% compared to HAp-TiN multilayer coating (0.06 μA/cm^2^).

## 6. Reinforcement of Sol-Gel Derived Hydroxyapatite by Composite Formation and Ion Exchange

### 6.1. Hydroxyapatite—Ceramic Systems

The studies of HAp-ceramic composites coatings have demonstrated the importance of ceramic particulate reinforcement in overcoming many limitation associated with pure HAp coating. In this sense, TiO_2_ has attracted considerable attention as a reinforcement to HAp coating, since it is biocompatible, has high chemical affinity towards Ti and HAp and its ability to improve the low strength of pure HAp coating. To confirm this, Kim et al. [121] deposited HAp and HAp-TiO_2_ composites with different molar percentage of TiO_2_ on Ti Substrate using sol-gel spin coating. They indicated higher roughness parameters in composite coatings and an increase in adhesive strength was observed with increasing TiO_2_ content. The highest strength was approximately 56 MPa with 30% TiO_2_ addition equivalent with an improvement of approximately 50% with respect to pure HAp coating. In another study, Han et al. [133] coated pure Ti with HAp-TiO_2_ composite using sol-gel spin coating followed by calcination at different temperatures. The XRD observations found that HAp start to crystallize at 550 °C and showed an improvement in crystallization and grain size growth at higher temperatures. However, delaying in crystallization and acceleration of decomposition of HAp is noted with the addition of TiO_2_ and this considered the main drawback of such addition. The adhesive strength of pure HAp and HAp-TiO_2_ composite was similar and relatively low at 450 °C, but with increasing temperature, the HAp composite showed higher improvement in bonding strength compared to the pure HAp. Based on potentiodynamic anodic polarization curves, an improvement in corrosion resistance is indicated with the addition of TiO_2_. Furthermore, Im et al. [92] examined the influence of the incorporation of TiO_2_ on the HAp coating, which was deposited on cp-Ti via a sol-gel spin coating. An increase in the surface roughness of the coatings from 0.831 to 0.969 μm is found with increasing content of TiO_2_. This is helpful to enhance the wettability and the interfacial bonding strength of the TiO_2_-HAp interface. The critical load for debonding of the hybrid HAp-TiO_2_ coating films to Ti substrate increased noticeably from below 2N for the pure HAp coating to over 5N for the high content TiO_2_ films. The bioactivity of the deposited films proved by the formation of needle-shaped bone-like apatite after 14 days of exposure to simulated body fluid. Dikici et al. [134] coated Ti alloy with HAp and HAp-TiO_2_ composite by sol-gel technique under different sintering parameters related to sintering temperatures and heating ramp speed. High heating ramp rate indicated to produce HAp coating with less porosity but higher bonding with the substrate and higher hardness. As summarized in Figure 6, the hardness of all the coated samples found to increase with increasing the sintering temperature from 600 °C to 800 °C and with TiO_2_ addition, the highest hardness value is noted with the composite coating sintered at 800 °C. In addition, the hardness increases with increasing the TiO_2_ until it reaches its maximum effect at 10 vol.%.

Choudhury and Agrawal [135] deposited films of HAp and HAp-ZrO_2_ composite on oxidized and passivated cp-Ti substrates by sol-gel route. Remarkable improvement in the interfacial shear strength of the coating/metal is observed after introduction of the ZrO_2_, while the oxidized samples found to have much lower interfacial strength than passivated ones. In spite of the higher coefficient of friction (COF) for the passivated samples compared to the oxidized ones, the passivated samples showed better wear resistance due to their creation of a rough surface that can promote the bonding with the HAp coating. In another study, Anjaneyulu and Vijayalakshmi [82] investigated the impact of incorporation of magnetite (Fe_3_O_4_) to sol-gel derived HAp coating deposited on alkali-treated Ti6Al4V alloy. The wettability of the composite coating have improved, which, in turn, promoted the bioactivity by increasing the formation rate of the bone-like apatite compared to bare HAp coating. In impedance and tafel polarization studies, the HAp-Fe_3_O_4_ composite showed lower corrosion current and higher corrosion resistance than bare HAp coated, and alkali-treated Ti6Al4V. However, incorporating 1 wt.% of Fe_3_O_4_ with HAp produced a uniform and adherent layer on the substrate and had a better corrosion behavior then the composites with higher Fe_3_O_4_ content.

### 6.2. Hydroxyapatite—Carbon Nanotube Systems

Multi-walled carbon nanotubes (MWCNTs) is an interesting material with potential biomedical application, many in-vivo and in-vitro studies have proven the high biocompatibility of the HAp-MWCNTs composites in the orthopedic applications [88,136]. Park et al. [89] indicated that the introduction of MWCNTs to sol-gel HAp coating on Ti can serve as a favorable location to form HAp nucleus, where the crystallization of the HAp coating observed to be enhanced with increasing the concentration of the MWCNTs up to 1 wt.% as shown in Figure 7. The same results were found by Park et al. [88]. In another study, Liu and Ji [137] successfully deposited dense and crack-free hydroxyapatite reinforced with MWCNTs on Ti substrate by sol-gel method. An improvement in the bonding strength of 22.2 MPa for the bare HAp coating to a maximum value of 32.9 MPa for HAp-1% MWCNTs is reported.

### 6.3. Substituted Hydroxyapatite Coating Systems

Infection is one of the most important causes of orthopedic prostheses failure, thousands of bone-substitute implants are revised every year due to the infections caused by different microbial as *S. epidermidis* and *S. aureus* [138]. The lack in the infection control and the poor antimicrobial property of the HAp is a serious drawback effecting on the long-term reliability of this coating. Fortunately, HAp is able to substitute the Ca^2+^ in its structure by other antibacterial metal ions such as Zn^2+^, Cu^2+^ and Ag^+^. When these metal ions reach the membrane of the microbial cells, they readily absorbed and penetrate into the cell wall causing lose in the proliferation capacity and death of the bacteria cells. Thus, incorporation of these ions and managing this substitution could result in new compounds with antibacterial property and without compromising the biological efficacy and bioactivity of the HAp matrix [139,140].

Jingying Zhang [141] doped Zn in the biological HAp coating and made the deposition on oxidized titanium by sol-gel method. The results observed 51.02% improvement in the inhibitory rate of *P. gingivalis* growth for Zn-HAp composite coating compared to bare HAp coating, and this improvement increase with increasing Zn concentration reaching up to 65.8% with 0.01 of Zn/Ca molar ratio.

Meanwhile, Batebi et al. [142] doped silver and fluoride with different percentages in HAp and coated the produced composite on titanium substrate by sol-gel method. Their results found that doping trace of Ag up to 0.3 wt.% has no effect on the HAp structure but could stimulate the formation of β-Tricalcium phosphate, while the presence of fluoride can resulting compact coating with finer crystallite sizes. The results of antibacterial test against *Escherichia coli* (*E. coli*) indicated that incorporation of Ag ions in the HAp structure promote the reduction in the E. coli cells number by 76% after 6 h. However, addition of fluoride ions in the Ag-HAp composite resulted more enhancement in reduction of bacteria number up to 96% after 6 h. In another recent study, Bi et al. [143] have fabricated biphasic coatings of Zn substituted HAp/Bismuth (Bi) substituted HAp (Zn-HAp/Bi-HAp) by sol-gel and deposited them on titanium substrate by dip-coating approach. The antibacterial behavior of the biphasic Zn-HAp/Bi-HAp coating against *S. aureus* and *Escheria coli* showed noticeable improvement with lower number and size of bacteria compared to the pure HAp coating. In addition, Zn-HAp coating promoted the chemical stability and decreased the dissolution rate of apatite in Tris buffer compared to biphasic and pure HAp coatings.

Fluoride ion (F-) is another interesting dopant which exists in the human bones as an important substance against the dissolution [144]. Doping fluoride in the HAp coating structure has drawn many attention in the recent studies due to its role in promoting crystallization, mineralization and reduction the dissolution rate of HAp coating which is make it more suitable for the bio-applications that require long term mechanical and chemical stability [55,144,145,146]. Zhang et al. [147] deposited crack-free and dense fluoridated hydroxyapatite (FHAp) film on Ti alloy by sol-gel. The bonding strength of the FHAp have improved over 35% compared to pure HAp and this improvement is found more prominent at elevated sintering temperature and is possibily due to the formation of chemical bonding at FHAp-Ti interface and incorporation of fluoride ions in HAp structure provided relief of thermal mismatch. In a similar study, Zhang et al. [148] reported significant improvement in biodegradation resistance and biocompatibility of the FHAp suggesting it as good replacement to pure HAp as coating on metallic implants. In addition, Tredwin et al. [149] reported a reduction in crystallization temperature and decreasing in cell size and viscosity of produced sol with incorporation fluoride in the HAp structure.

Other types of substitution elements doped in HAp coating structure and prepared by other techniques were investigated such as zinc, copper, silver, magnesium, strontium cobalt etc. Arcos and Vallet-Regí [138] have well discussed in their review the important of these dopants in promoting the HAp coating function.

### 6.4. Hydroxyapatite—Polymer System

The HAp-polymer composite coatings have a unique combination of mechanical and biological properties making them noteworthy for many biomedical applications. One of the main advantages of polymer addition is lowering the stiffness and elastic modulus of HAp that limits the stress-shielding at the bone-HAp interface. However, poor adhesion strength is the main drawback of this system due to the high oxygen ratio in the molecular structure of the polymers that supply more electrostatic-interactions on the surface of the substrate. Various types of polymers are used as additives with HAp such as polycaprolactone, collagen, chitosan, polyethylene glycol etc. [150]. Polycaprolactone (PCL) has attracted much attention as a promising additive to HAp due to its good mechanical properties and biocompatibility. In this sense, a study made by Mohd et al. [91] have been done to investigate the influence of the PCL addition with different concentration on HAp coating synthesised by sol-gel and deposited on a Ti6Al4V substrate. As demonstrated in Figure 8, very thin and loosely packed layers are observed for the pure HAp coatings. However, introduction of PCL up to 30% and 50% resulting thicker, densely packed, porous and crack-free coatings with enhanced cohesion and adhesion to the Ti substrate. Furthermore, the addition of PCL decreased the corrosion current and improved the corrosion resistance, lowest corrosion current was found with HAp-50% PCL sample. Similarly, Ansari et al. [151] confirmed the active role of PCL in improving the biological, physical and electrochemical properties of FHAp coatings. Contrary to Mohd et al. investigation, the PCL considered as a matrix and consist up to 30 wt.% of FHAp, and the resulted composite deposited on alkali-treated Ti alloy via sol-gel dip-coating. Improvement in surface roughness is reported after introduction of FHAp, while the bioactivity verification test in SBF has confirmed the satisfactory apatite formation ability of the composites with high FHAp content. The corrosion resistance is remarkably enhanced after the PCL-FHAp composite deposition and its much noticeable in PCL-10 wt.% FHAp compared to other composites. However, poor adhesion strength of all the composite coatings are observed where the maximum strength achieved was 3.08 MPa for the PCL-30 wt.% FHAp film.

## 7. Summary and Future Perspectives

Surface modifications of the Ti-based biomaterials have become a vital topic, especially to overcome their instability and promote their biocompatibility and bioactivity. Sol-gel technique for deposition of hydroxyapatite is continuously being investigated as a potential technique to overcome the limitations of Ti implants. This review has demonstrated the advantages and limitations of deposition HAp by sol-gel and surveyed the developments of this coating. In this review, it was clearly highlighted that the poor adhesion of the pure HAp to the Ti surface and the low corrosion resistant of the coated Ti were the main drawbacks. Table 4 represents a summary of the identified adhesion and corrosion resistance improvements of sol-gel HAp coating on Ti substrates.

Several studies showed the importance of some sol-gel parameters on the structural, morphological, corrosion and mechanical properties of the HAp coatings. However, there are still some deficiencies, altering the initial precursors, conditions of sol preparation, gelation time, physical processing or condition of gelation of the gel itself can control the performance of the HAp coating [152]. Thus, more extensive investigations to optimize the sol-gel parameters is important to achieve ideal HAp coating for orthopedic applications. Meanwhile, other studies showed that pretreatments of the Ti substrate surface by formation of interlayers e.g., TiO_2_, TiB or TiN are tightly adherent to HAp and Ti surface and positively affecting on structural, corrosion and mechanical properties of the sol-gel hydroxyapatite coating. However, more systematic efforts are needed to improve the interlayer benefits by controlling its properties, particularly its thickness and surface roughness. The thickness of the interlayer is directly effecting on the bonding strength of the double layer and the corrosion resistance of the Ti substrate. Thick interlayer promotes the chemical stability of the Ti substrate by being as a barrier against corrosion but deteriorate the bonding strength of the multilayer due to the mismatch in the CTE between the intermediate layer and Ti substrate, and vice versa. Thus controlling the intermediary layer thickness to compromise between corrosion resistance and interfacial strength is important. In addition, rough surface of interlayer is beneficial as mentioned previously by Xu et al. [127] due to its capability to enhance the wettability of the HAp coating and improve the adhesion-based mechanical interlocking theory. Thus, controlling the parameters of interlayer manufacturing process to achieve optimized value of roughness is essential. On the other hand, HAp-based composites have been the focus in many studies and showed a significant potential to improve the biochemical and mechanical properties compared to pure HAp coating. It can be noted a considerable demand for a detailed evaluation of the effective factors of incorporated fillers and ions in enhancing the final properties of HAp-based composite coatings.

Altogether, sol-gel derived HAp-based composites and multilayer structures synthesized with optimized processing parameters have bright prospect due to their benefits from biocompatibility as well as enhanced the corrosion resistant and the adhesion to the Ti substrate. However, completion of the correlation knowledge between the sol-gel processing parameters and the final properties of the coating to improve the physical, electrochemical, biological, and mechanical properties of the coating, is still a challenge for sol-gel HAp coatings for Ti-based implants.

## Figures and Tables

**Figure 1 bioengineering-07-00127-f001:**
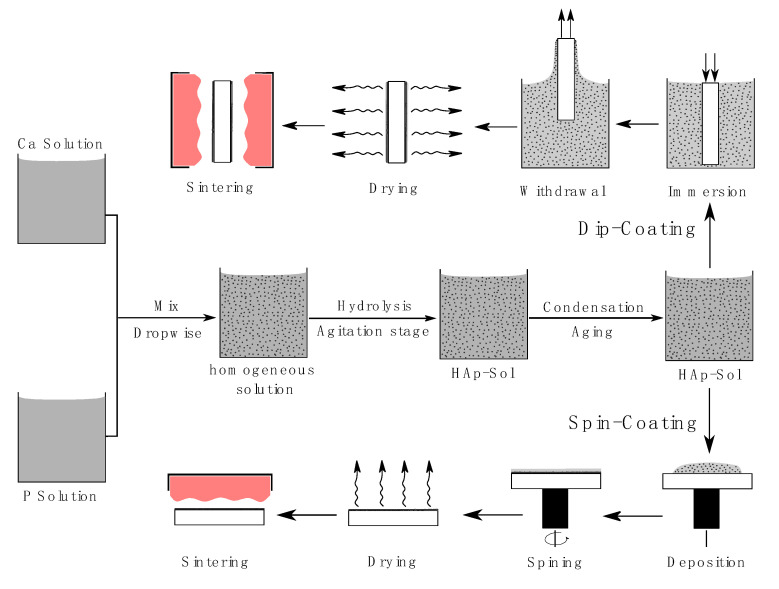
Fundamental stages of sol-gel HAp preparation and deposition by dip and spin coating.

**Figure 2 bioengineering-07-00127-f002:**
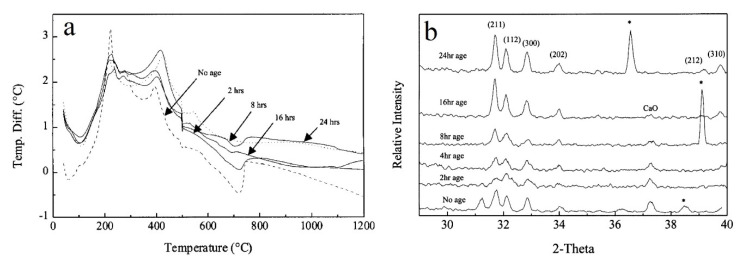
(**a**): DTA and (**b**): XRD for HAp coating aged up to 24 h [66], reprinted with permission from Elsevier.

**Figure 3 bioengineering-07-00127-f003:**
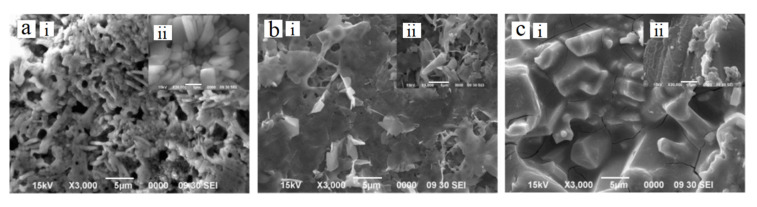
HAp coating on Ti alloy sintered at (**a**): 600 °C, (**b**): 700 °C and (**c**): 800 °C. Note: (i,ii) indicate different magnification of the micrographs [117].

**Figure 4 bioengineering-07-00127-f004:**
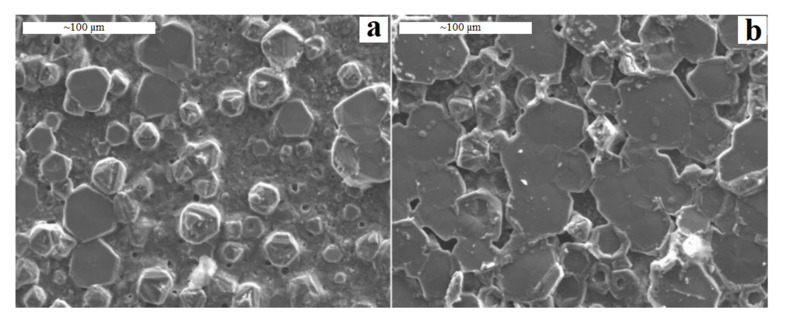
SEM images of HAp coating dried at 500 °C and calcined at 800 °C with (**a**); (**a**): rapid heating rate and (**b**): slow heating rate [118], reprinted with permission from Taylor & Francis Ltd.

**Figure 5 bioengineering-07-00127-f005:**
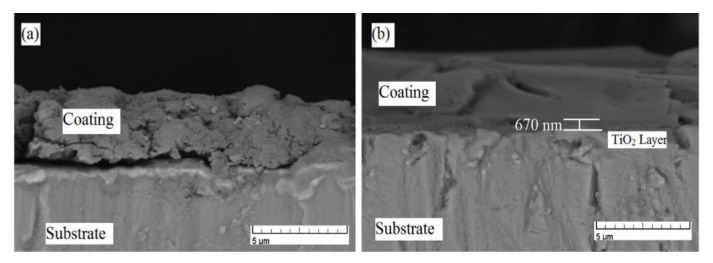
Cross-sectional SEM images of (**a**) HAp and (**b**) FGC HAp-TiO_2_ coatings [33], reprinted with permission from Elsevier.

**Figure 6 bioengineering-07-00127-f006:**
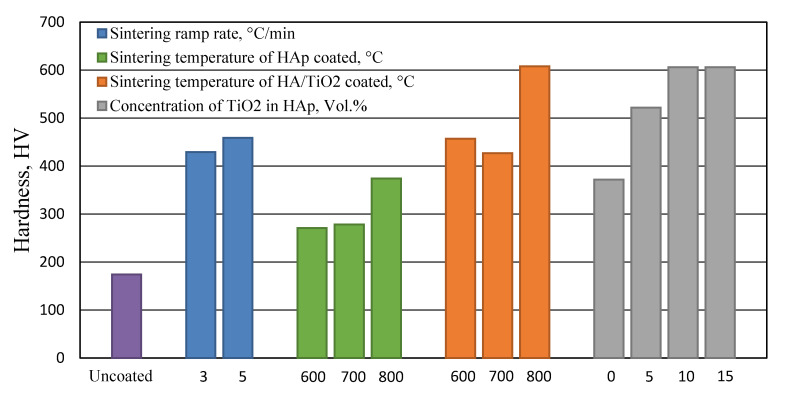
Effect of TiO_2_ addition and sintering parameters on the coating hardness (Data from [134]).

**Figure 7 bioengineering-07-00127-f007:**
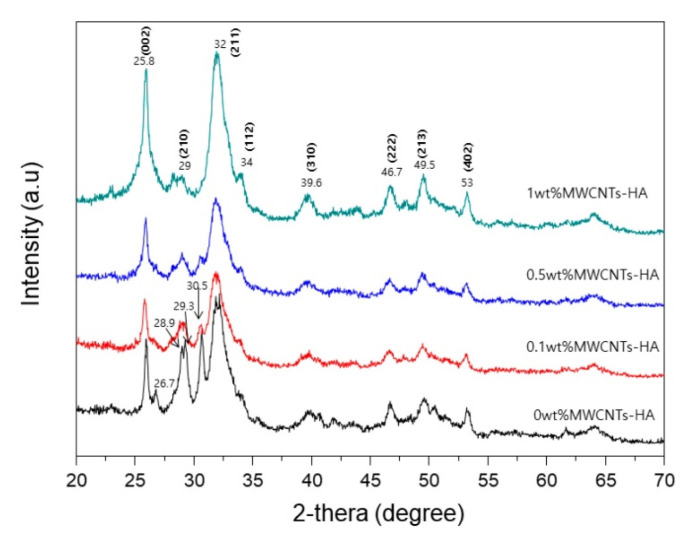
XRD patterns of HAp-MWCNTs composites [89].

**Figure 8 bioengineering-07-00127-f008:**
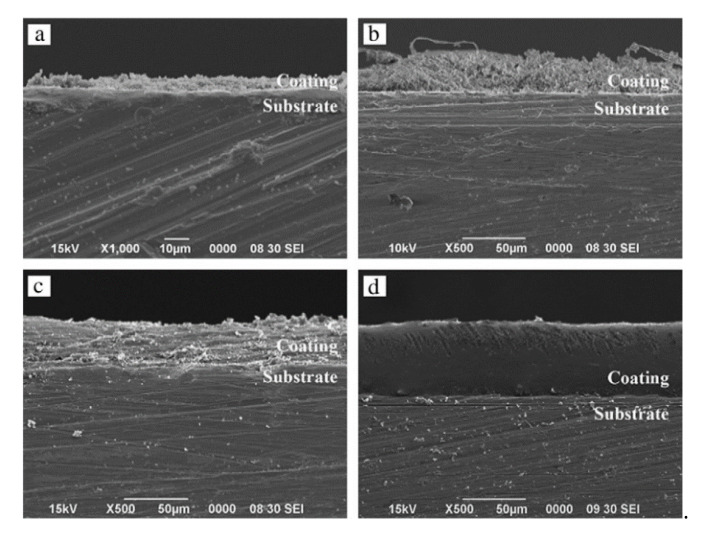
SEM images of the cross-section of (**a**) HA, (**b**) 10% PCL/HA, (**c**) 30% PCL/HA, and (**d**) 50% PCL/HA [91], reprinted with permission from Elsevier.

**Table 1 bioengineering-07-00127-t001:** Hydroxyapatite (HAp) coating requirements [27].

Property	Specification
Thickness	Not specified
Crystallinity	62% minimum
Phase purity	95% minimum
Ca/P ratio	1.67–1.76
Density	2.98 g/cm^3^
Heavy metals	<50 ppm
Tensile strength	>50.8 MPa
Shear strength	>22 MPa
Abrasion	Not specified

**Table 2 bioengineering-07-00127-t002:** Different deposition techniques of HAp coatings.

Approach	Thickness	Benefits	Weaknesses	References
Sol–gel	<1 µm	Low processing temperature, inexpensive and produce pure and very thin coating on flat and complex shapes.	Poor adhesion to substrate and controlled atmosphere is required in some stages.	[48,49,50]
Plasma spraying	<20 µm	Lower possibility of coating degradation, rapid deposition and in expensive method.	Non-uniform coating density, poor adhesion and high processing temperature causes decomposition of HAp and phase transformation of substrate.	[51,52,53,54]
Electrophoretic deposition	100–2000 µm	Coating complex shapes, uniform coating thickness and high deposition rate.	High processing temperature and difficulty in producing crack-free coating	[25,55,56,57]
Electrochemical deposition	50–500 µm	Rapid deposition, inexpensive, coating complex shapes and produce uniform coating thickness	Poor adhesion quality between coating and substrate	[58,59,60]
Biomimetic coating	<30 µm	Coating complex shapes, low processing temperature and able to form bone-like apatite	Slow deposition rate and produce low degree of crystallinity	[61,62,63]

**Table 3 bioengineering-07-00127-t003:** Precursors and solvent of hydroxyapatite.

	Precursor	Solvent	References
**Ca Precursors**	Calcium acetate monohydrate	Water and 1,2-ethanediol	[78,79,80]
	Calcium nitrate tetrahydrate	Water	[33,81]
	Calcium nitrate tetrahydrate	Ethanol	[47,82,83,84,85,86]
	Calcium nitrate tetrahydrate	Water and ethanol	[32,87]
	calcium chloride	Water	[31]
**P Precursors**	Phosphoric acid	Water	[80]
	Ammonium phosphate dibasic	Water	[33,81,88,89,90,91]
	Triethyl phosphite	Water and ethanol	[92,93,94,95,96]
	Triethyl phosphite	Ethanol	[82]
	Trimethyl phosphate	Water and ethanol	[32]
	Phosphorus pentoxide	Ethanol	[47,85,86,97]
	diammonium hydrogen orthophosphate	Water	[84]
	Trisodium phosphate	Water	[31]

**Table 4 bioengineering-07-00127-t004:** Summary and comparison of the adhesion and corrosion resistance improvements of sol-gel derived HAp coatings on Ti substrates.

**Attempts to Improve the Adhesion of Sol-Gel Hydroxyapatite Coating on Ti Substrate.**								
**No.**	**Type of Improvement**	**Substrate**	**Adhesion of Pure HAp**		**Adhesion after Improvement**	**Type of Test**	**Sintering Temperature (°C)**	**Ref.**
1	TiO_2_ interlayer prepared by sol-gel	cp Ti	35 MPa		55 MPa	Pull-out test	500	[125]
		Ti6Al4V	15.8 ± 7 MPa		40.3 ± 3 MPa	Pull-out test	600	[126]
2	TiO_2_ interlayer prepared by anodization	cp Ti	538 ± 40 MPa		652 ± 12 MPa	Micro-tensile test	550	[130]
		Ti6Al4V	1073 ± 30 MPa		1086 ± 40 MPa	Micro-tensile test		[130]
		Ti6Al4V	13.8 ± 3.28 MPa		19.02 ± 3.36 MPa	Pull-out test	600	[93]
3	TiO_2_ interlayer prepared by anodization -HAp-1% MWCNTs	cp Ti	35.2 MPa		21.2 MPa	Pull-out test	550	[82]
4	TiB and TiB_2_ interlayers prepared by boronisation	cp Ti	3 MPa		15 MPa	Pull-out test	400	[90]
5	HAp-30 mol.% TiO_2_ composite	cp Ti	37 MPa		70 MPa	Pull-out test	500	[121]
6	HAp-20 mol.% TiO_2_ composite	cp Ti	34 MPa		50 MPa	Pull-out test	750	[133]
7	HAp-1 wt.% MWCNTs composite	cp Ti	22.2 MPa		32.9 MPa	Pull-out test	550	[137]
8	HAp-ZrO_2_ composite	cp Ti	570 MPa		678 MPa	lag-shear strain method	700	[135]
**Attempts to Improve Corrosion Resistance of Ti Substrate Coated by Sol-Gel Hydroxyapatite**								
**No.**	**Type of Improvement**	**Substrate**	**C.R Indicator**	**C.R of Substrate**	**C.R of Pure HAp**	**C.R after Improvement**	**Sintering Temperature (°C)**	**Ref.**
1	TiO_2_ interlayer applied by sol-gel	cp Ti	I_corr_ (A/cm^2^)	~ 4.2 × 10^−5^	~ 9.5 × 10^−7^	N/A	500	[125]
		Ti-15Zr-12Nb	Corrosion resistance (mpy)	2.3104	N/A	0.8633	550	[128]
2	TiO_2_ interlayer applied by anodization	cp Ti	I_corr_ (A/cm^2^)	N/A	N/A	4.05 × 10^−7^	N/A	[131]
		cp Ti	E_corr_ (V)	−1.204	N/A	−0.517		
3	TiN interlayer applied by PACVD	Ti6Al4V	I_corr_ (A/cm^2^)	1.17 × 10^−6^	1.2 × 10^−7^	0.6 × 10^−7^		
4	HAp-20 mol.% TiO_2_ Composite	cp Ti	E_corr_ (V)	N/A	−0.36	−0.324	750	[133]
5	HAp–1 wt.% Fe_3_O_4_ Composite	Ti6Al4V	Corrosion resistance (mpy)	0.286 ± 0.011	N/A	0.059 ± 0.007	800	[82]
6	HAp–50 wt.% PCL Composite	Ti6Al4V	I_corr_ (A/cm^2^)	3.3497 × 10^−7^	6.5257 × 10^−8^	9.3478 × 10^−9^	No sintering	[91]
7	HAp–90% PCL Composite	Ti6Al4V	I_corr_ (A/cm^2^)	N/A	N/A	1.41 × 10^−9^	No sintering	[151]
8	Optimization of sintering temperature	Ti13.5Zr14Nb	I_corr_ (A/cm^2^)	3.1 × 10^−5^		1.6 × 10^−5^	500	[31]
						6.2 × 10^−6^	600	
						1.8 × 10^−6^	700	

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
