# Peer review of "Sol-Gel Derived Hydroxyapatite Coatings for Titanium Implants: A Review"

_bioengineering, 2020, doi:10.3390/bioengineering7040127_

Round 1
Reviewer 1 Report
Dear authors,
The manuscript is very well written and contains the comprehensive review which is of remarkable importance for improving Ti-based bone implants for their better osteointegration. The sections of the manuscript are well organized and conclusions and outlooks are clearly presented and explained.
Please consider the following suggestion:
Page 8, line 280-281.
The statement about better implant fixation due to formation of the fibrous capsule around the implant is unclear, considering its undesirable effect on the final osteointegration of implant. It is known, that the fibrous capsule around the implant might prevent the establishing the interface between implant surface and host bone, and therefore, interfere the osteointegration of the implant (For example, it was described in details in this study https://doi.org/10.1016/S0142-9612(99)00151-9). Please check this sentence at lines 280-281.
Author Response
Comment 1. Page 8, line 280-281.
The statement about better implant fixation due to formation of the fibrous capsule around the implant is unclear, considering its undesirable effect on the final osteointegration of implant. It is known, that the fibrous capsule around the implant might prevent the establishing the interface between implant surface and host bone, and therefore, interfere the osteointegration of the implant (For example, it was described in details in this study https://doi.org/10.1016/S0142-9612(99)00151-9). Please check this sentence at lines 280-281.
Reply: Minimal encapsulation assist in biological implant fixation, and as mentioned in the manuscript, one of the important advantages of introduction the interlayers is to improve the implant fixation due to the minimal fibrous capsule around the implant.
Reviewer 2 Report
The authors summarised the sol-gel technique for micro-scaled HA coating for Ti implants. The reviewer has a few concerns: 1. Number Formating for each section is always 1, which is confusing. 2. From the abstract, the reviewer cannot see why this abstract is needed. The authors should focus on why this abstract is required for the field. 3. In vitro and in vivo should be italic throughout the whole manuscript. 4. All the characterisation seem to be SEM and XRD, which is lacking the detailed methods for characterisation. The authors should include other methods. Also, have the authors obtained permission to use the figures? 5. The authors mentioned that sol-gel techniques can lead toAuthor Response
Comment 1. Number Formating for each section is always 1, which is confusing.
Reply: the numbering of the sections is corrected.
Comment 2. From the abstract, the reviewer cannot see why this abstract is needed. The authors should focus on why this abstract is required for the field.
Reply: The abstract is modified (line 13-16).
Comment 3. In vitro and in vivo should be italic throughout the whole manuscript.
Reply: the terms have been reformatted throughout the text.
Comment 4. All the characterisation seem to be SEM and XRD, which is lacking the detailed methods for characterisation. The authors should include other methods. Also, have the authors obtained permission to use the figures?
Reply: In addition to SEM and XRD characterization, some other important properties of the hydroxyapatite coatings (i.e. adhesive strength, corrosion resistance, bioactivity etc.) were focused on and discussed. The permissions to use the figures are granted and submitted to the journal.
Reviewer 3 Report
The review article presented for peer-review concerns the hydroxyapatite coatings produced by so-gel method. The topics discussed are up-to-date and very important. In my opinion, such a review article is needed to collate knowledge on such coatings.
The presented work fits well with the topic of Bioengineering journal.
Minor comments:
- No purpose of work in the introduction section
- Several spaces are missing between the text and the quotation brackets (for example line 22, 35 etc.)
- Something is missed in the sentence starting with “Thus, there is a great for appropriate …” (line 22)
- “High specific strength” is related to density as well as “relatively low weight” (lines 34-35). I think in this case you only need to use "strength"
- “Mechanical properties are limited” (line 39) – which properties ?
- Only the positive aspects of titanium alloys with the beta structure are shown. It is suggested to supplement with their disadvantages (since alloys with α+β structure are preferred)
- Ti6Al4V alloy was indicated as the most preferred. And it has been known for a long time that both Al and V are toxic or otherwise undesirable. I suggest briefly describing this problem, pointing to new titanium alloys devoid of the above-mentioned disadvantages. A review article may be helpful here: Palka K., Pokrowiecki R.: Porous titanium implants: a review. Advanced Engineering Materials.- 2018, vol. 20, nr 5, s. 1-18 https://doi.org/10.1002/adem.201700648
- Sentence starting with “The bioinert ceramic coatings…” (lines 70-72) seems to be incomprehensible.
- In the paragraph starting at line 75, the authors focused on hydroxyapatite. I suggest a brief mention of the other calcium phosphates.
- Chapter numbering should be corrected
- Chaper 2 – Deposition techniques - there is no justification as to why the authors focused on the sol-gel method.
- Chapter 3.3: The title can be shortened by removing the word “Known”
- Figure 3 must be improved – scale bars in insets are invisible.
Author Response
Comment 1. No purpose of work in the introduction section.
Reply: The purpose of work is presented (lines 187-188)
Comment 2. Several spaces are missing between the text and the quotation brackets (for example line 22, 35 etc.).
Reply: spaces are added throughout the manuscript.
Comment 3. Something is missed in the sentence starting with “Thus, there is a great for appropriate …” (line 22).
Reply: the sentence is modified “Thus, there is a great demand for appropriate …” (line 25).
Comment 4. “High specific strength” is related to density as well as “relatively low weight” (lines 34-35). I think in this case you only need to use "strength".
Reply: the sentence is modified, the word ‘specific’ is removed in line 35.
Comment 5. “Mechanical properties are limited” (line 39) – which properties ?
Reply:The low tensile strengths of the alpha titanium alloy in comparison to the alpha-beta alloys is the main limited mechanical property of this kind of alloys.
Comment 6. Only the positive aspects of titanium alloys with the beta structure are shown. It is suggested to supplement with their disadvantages (since alloys with α+β structure are preferred).
Reply: The main limitation of beta alloys is their low corrosion resistance compared to the alpha alloys, added to the manuscript, line 40-41.
Comment 7. Ti6Al4V alloy was indicated as the most preferred. And it has been known for a long time that both Al and V are toxic or otherwise undesirable. I suggest briefly describing this problem, pointing to new titanium alloys devoid of the above-mentioned disadvantages. A review article may be helpful here: Palka K., Pokrowiecki R.: Porous titanium implants: a review. Advanced Engineering Materials.- 2018, vol. 20, nr 5, s. 1-18 https://doi.org/10.1002/adem.201700648.
Reply: Brief description on the new generations of biomedical titanium alloys are presented (lines 46-50)
Comment 8. Sentence starting with “The bioinert ceramic coatings…” (lines 70-72) seems to be incomprehensible.
Reply: The paragraph has been extended to better describe the bioactive and bioinert ceramic coatings (lines 74-80)
Comment 9. In the paragraph starting at line 75, the authors focused on hydroxyapatite. I suggest a brief mention of the other calcium phosphates.
Reply: Introduction to calcium phosphate and its types is presented (lines 81-85)
Comment 10. Chapter numbering should be corrected.
Reply: the numbering of the sections is corrected.
Comment 11. Chaper 2 – Deposition techniques - there is no justification as to why the authors focused on the sol-gel method.
Reply: based on the chapter 2 and table 2, most of the coating techniques, especially plasma spraying, have the high temperature problems that could cause decomposing of hydroxyapatite and forming undesirable phases, in addition to their high coating thickness. While as mentioned in chapter 3, the sol-gel method have many advantages to overcome the limitations of other techniques.
Comment 12. Chapter 3.3: The title can be shortened by removing the word “Known”
Reply: modified as suggested (the word ‘Known’ is removed – line 161).
Comment 13. Figure 3 must be improved – scale bars in insets are invisible.
Reply: The scale bars in figure 3 is improved to be clearer as suggested.